# United States National Postdoc Survey results and the interaction of gender, career choice and mentor impact

**Abstract** The postdoctoral community is an essential component of the academic and scientific workforce, but a lack of data about this community has made it difficult to develop policies to address concerns about salaries, working conditions, diversity and career development, and to evaluate the impact of existing policies. Here we present comprehensive survey results from 7,603 postdocs based at 351 US academic and non-academic (e.g. hospital, industry and government lab) institutions in 2016. In addition to demographic and salary information, we present multivariate analyses on factors influencing postdoc career plans and satisfaction with mentorship. We further analyze gender dynamics and expose wage disparities. Academic research positions remain the predominant career choice, although women and US citizens are less likely than their male and non-US citizen counterparts to choose academic research positions. Receiving mentorship training has a significant positive effect on postdoc satisfaction with mentorship. Quality of and satisfaction with postdoc mentorship also appear to heavily influence career choice.
DOI: https://doi.org/10.7554/eLife.40189.001

**SEAN C MCCONNELL[†], ERICA L WESTERMAN[†*], JOSEPH F PIERRE, ERIN J HECKLER AND NANCY B SCHWARTZ**

*For correspondence: ewesterm@uark.edu

[†]These authors contributed equally to this work

Competing interests: The authors declare that no competing interests exist.

## Introduction

Postdoctoral training offers doctoral recipients a temporary period of mentored or scholarly experience, considered highly productive within scientific and academic communities. Such training is also ostensibly valuable for postdocs, who gain additional experience to help pursue their chosen career paths. Tenure-track faculty positions, however, are now estimated to represent a small percentage of postdoc career outcomes (~15%) (*Larson et al., 2014*; *National Academy of Sciences, 2014*). This has led to proposals that support training postdocs for additional roles beyond tenure-track faculty positions. Additional efforts by the National Institutes of Health (NIH), National Science Foundation (NSF), and National Academies of Sciences, Engineering, and Medicine also aim to increase mentor accountability (*National Academy of Sciences, 2014*; *Institute of Medicine, 2000*;

*The National Academies of Sciences, 2018*; *National Institutes of Health, 2012*; *Meyers et al., 2016*). Persistent concerns with increasingly long periods of postdoctoral training, lack of appropriate career guidance beyond the professoriate, and comparatively low postdoctoral salaries have also led to repeated calls to reform the postdoctoral training model (*National Research Council, 1969*; *Davis, 2005*; *Sauermann and Roach, 2016*; *Alberts et al., 2014*; *Gould, 2015*; *Schaller et al., 2017*).

Despite these concerns, comprehensive data for postdocs are not routinely collected (*The National Academies of Sciences, 2018*). Indeed, reliable data on such basic information as the number of postdocs have been lacking, or disputed, in part due to the difficulties that a lack of job title standardization, postdoc mobility, and the ad hoc nature of institutional

postdoctoral administration present to data collection efforts (*National Academy of Sciences, 2014*; *National Institutes of Health, 2012*; *Schaller et al., 2017*; *Daniels, 2015*; *Pickett et al., 2017*).

Possibly for these very reasons, the postdoctoral experience has not been comprehensively surveyed nationally in over a decade, following the Sigma Xi 'Doctors without orders' survey report in 2005, which was based on postdoc respondents from 46 participating institutions (*Davis, 2005*). Nevertheless, recent data collection efforts have provided insights into the postdoctoral experience (*National Institutes of Health, 2012*; *Sauermann and Roach, 2016*; *Pickett et al., 2017*; *Heggeness et al., 2016*; *Gibbs et al., 2015*; *Ferguson et al., 2014*; *Phou, 2017*; *Kahn and Ginther, 2017*). For example, the pilot phase of the NSF Early Career Doctorates Survey studied the breadth of the doctoral population at US academic institutions, including postdoctoral researchers, early career faculty, and scientists in non-postdoctoral positions such as staff scientist or administrative positions (*Phou, 2017*). Most of their respondents were full-time faculty (54%); only 31% of respondents were postdocs.

Comprehensive survey data that specifically targets the postdoctoral period and includes postdoctoral researchers with PhDs granted both inside and outside the US, and data regarding postdoc career plans, satisfaction with mentorship, or family demographics, are still largely lacking (*National Academy of Sciences, 2014*; *The National Academies of Sciences, 2018*). To address these gaps, and to research those postdocs who do not have clear institutional oversight, we took a grass-roots approach to conduct a postdoc-led survey of US postdoctoral researchers. We asked postdoctoral researchers a number of questions associated with professional and career development, mentoring, career choice, lifestyle, and demographics (for details see Materials and Methods and *Source Data 1-3*). The purpose of this work was to capture a comprehensive snapshot of the postdoctoral experience in a manner that was both broad and informative, with high diversity in the questions and topics covered, in the number and type of institutions where the postdoctoral researchers are based, and in the breadth of postdoctoral experiences included.

## Results and Discussion

To collect data from institutions with a wide range of support for postdocs, we took a multi-level approach to recruit survey participants. We used publicly available contact information for university leadership, postdoctoral administrators, postdoctoral societies and associations, and asked these individuals in leadership positions to disseminate our survey to all postdocs at their institutions. In total, we contacted individuals at the 482 US institutions most likely to have postdocs, including universities, research institutions, museums and government labs. We obtained respondents from 351 institutions. In addition to direct contact with institutions, we also used a grass-roots survey dissemination approach, promoted a website describing the survey that could be freely shared on social media and by email, and contacted professional societies to encourage survey dissemination. Using these combined approaches, we collected 7,673 individual responses into a secure REDCap database (IRB Protocol Number 15–1724), which, after quality control to remove respondents from non-US institutions, provided a final dataset of 7,603 respondents (see Materials and Methods for details).

As one of our goals was to reach as many postdocs as possible, survey dissemination was not randomized to any specific subset. Responses from institutions with long-standing postdoctoral affairs offices were anticipated to be over-represented in our dataset (see Materials and Methods for more information). Nevertheless, our respondents represented all 50 states, including a large fraction of respondents from institutions without well-established offices for postdoctoral support. While the majority of respondents represented STEM disciplines, which traditionally employ the most postdocs, 8.4% reported their primary fields as humanities, psychology or social sciences (*Supplementary file 1* (Table S3)).

Our postdoc respondents were 49% US citizens and 51% non-US citizens (*Figure 1* and *Supplementary file 1* (Table S2)). The majority were 30–34 years of age (54.5%), and 1–3 years from receipt of their doctorate (63.1%), matching their reported years of postdoctoral experience (*Figure 1* and *Supplementary file 1* (Table S2)). The majority of postdoc respondents (69%) were from R1 academic institutions (Carnegie classification), with the remainder from non-R1

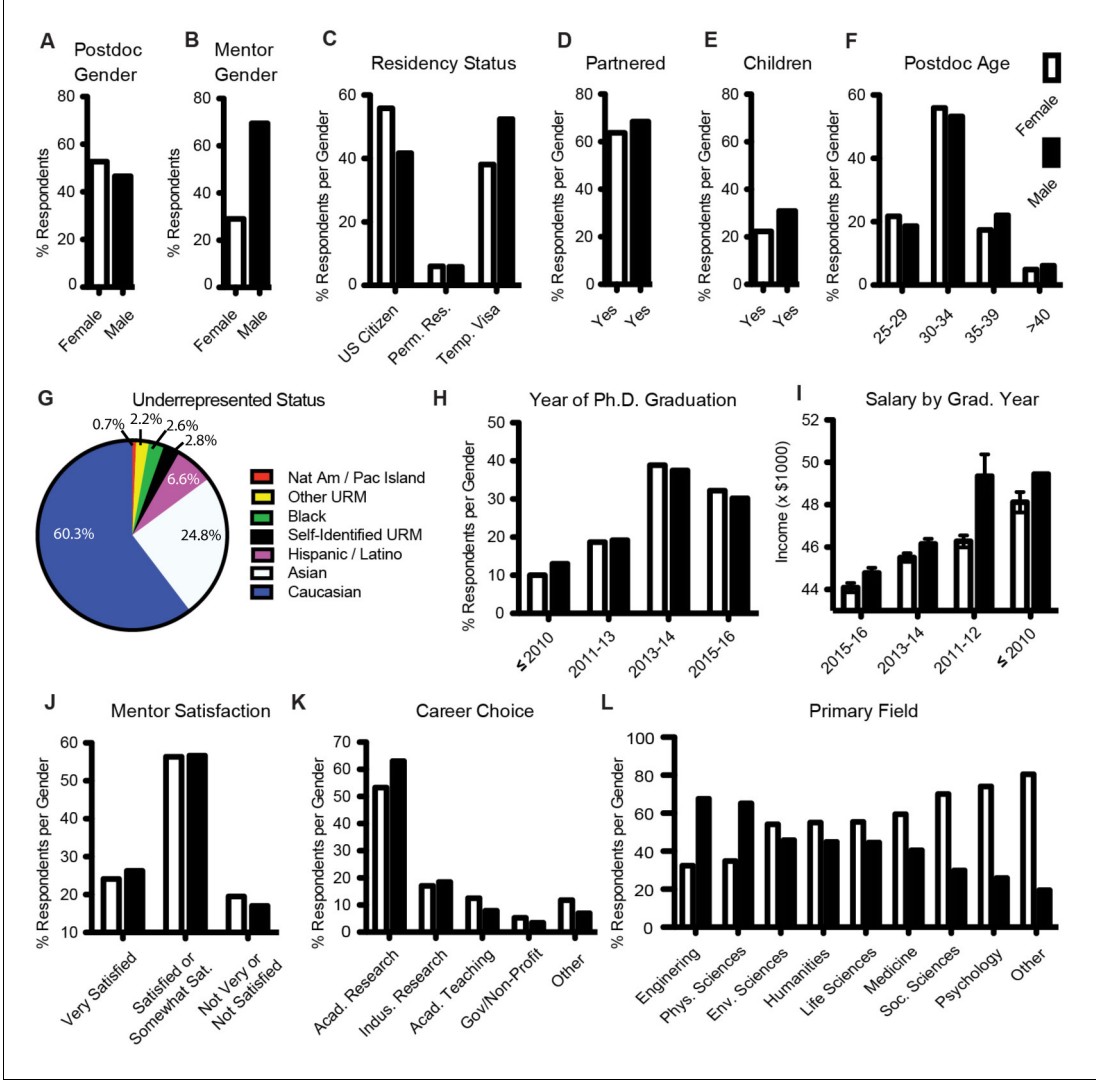

**Figure 1.** Demographics of the postdoc population surveyed. (**A**) Postdoc gender; (**B**) Mentor gender; (**C**) Residency status; (**D**) Partnered/Married; (**E**) Has children; (**F**) Age; (**G**) Race/Ethnicity/Underrepresented status (which may include things other than race and ethnicity, such as LGBTQ or disability status); (**H**) Year of graduation; (**I**) Adjusted income, by year of graduation; (**J**) Postdoc satisfaction with mentor; (**K**) Primary long-term career plans; and (**L**) Primary field/discipline. White bars indicate female, black bars indicate male.
DOI: https://doi.org/10.7554/eLife.40189.002

academic institutions (16%), medical centers (10%), and national government laboratories (4%). Less than 1% were either self- or otherwise employed. Of the 6,476 postdoc respondents in academia, 57% were based at public institutions while 43% were at private institutions. While the majority of postdocs were from R1 institutions, a large number of both R1 and non-R1 institutions were represented in our data set. Respondents were based at 151 different R1 institutions, 135 non-R1 academic institutions, 42 medical centers, and 59 national government laboratories. The Carnegie classification of non-R1 academic institution includes a wide range of academic institutions, from medical and PhD degree granting institutions that do not qualify as R1, to liberal arts and historically minority serving institutions. Due to the relatively small percentage of postdoc respondents from the large number of non-R1 institutions represented in our data set, we did not parse this group further, though future work should assess whether postdocs at liberal arts and minority serving institutes have unique experiences relative to postdocs based at other non-R1 academic institutions.

A majority of respondents (55%) described their primary field of study as life sciences. There were small, but significant, differences in primary field by geographic region (*Figure 2*; *Supplementary file 1* (Table S3)). Race and ethnicity were self-reported with 60.3% White/Caucasian, 24.8% Asian/Asian American, 6.6% Hispanic/Latino, and 2.6% Black/African American (*Supplementary file 1* (Table S2)). Both national and international postdocs were included in these proportions. Our respondents were 53% female, while the gender ratio of their mentors was skewed towards males (71% male; *Figure 1*), consistent with the most recent AAUP Gender Equity report where full-time faculty are majority (61%) male (*Association of American University Professors, 2006*).

While the demographics of our survey respondents may differ slightly from those of the actual postdoctoral population (but see Materials and Methods for analysis suggesting lack of response bias), confirmation of a lack of

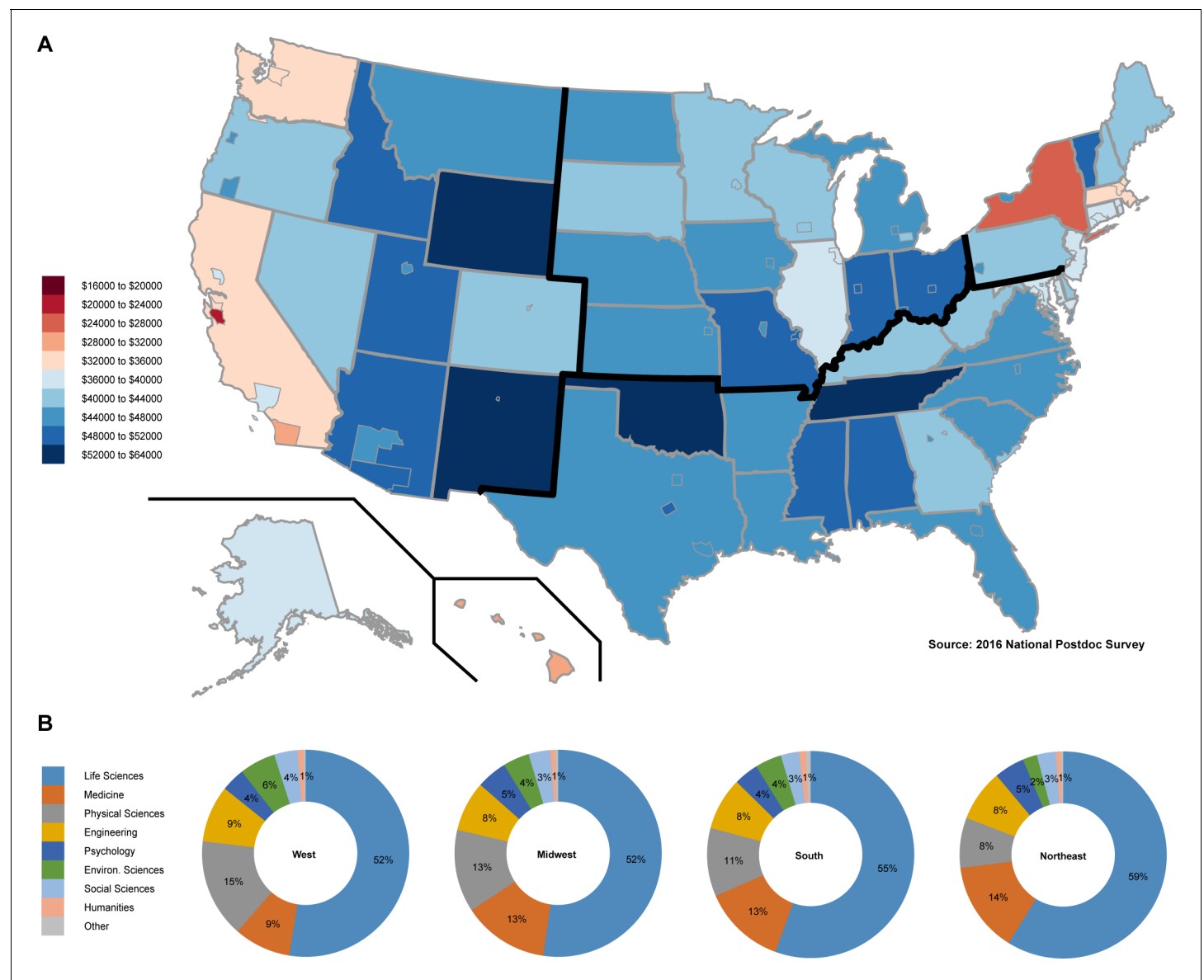

**Figure 2.** Postdoc cost of living adjusted income and field of study by region. (**A**) A map of the United States with the range of reported postdoc gross income adjusted by cost of living (key on the left). The adjusted income data are provided at the state (and when data sufficient to support, county) level. (**B**) The respondents' field of study (key on the right) in each of the four major regions: West, Midwest, South, and Northeast (designated by bold lines on the map in **A**).
DOI: https://doi.org/10.7554/eLife.40189.003

response bias remains difficult as there are currently no gold-standard datasets of postdocs in the US for comparison, due to the previously mentioned broader lack of oversight and barriers to reaching postdocs. That being said, the unique characteristics of our dataset, including approximately equal representation (similar sample sizes) of men and women, as well as US citizens and non-US citizens, facilitated our comparative analyses of the postdoctoral experiences of these different groups, which we report below.

Our data indicate that gender has a significant effect on the postdoc experience (*Figure 1*). Men were paid more than women (Male average: $47,678.00, Female average: $46,477.43, n = 7,516, $\chi^2$ = 62.337, p < 0.0001. Note that our survey answer options were categorical instead of continuous, thus Pearson $\chi^2$ tests and nominal logistic regression models were used instead of ANOVAs and generalized linear regression models throughout the manuscript). Men were more likely to have a same-gender mentor, i.e. a same-gender role model (Male: 77.3%, Female: 35.4%, n = 7,459, $\chi^2$ = 144.352, p < 0.0001). Men were more likely to be non-US citizens (Male: 42% US citizens, 52% temporary visas, 6% permanent residents; Female: 56% US citizens, 38% temporary visas, 6% permanent residents; n = 7,543, $\chi^2$ = 169.709, p < 0.0001). In addition, a small but significantly higher proportion of male postdocs were married/partnered (Male: 68.3%, Female: 63.2%, n = 7,538, $\chi^2$ = 21.693, p < 0.0001) and/or have children (Male: 31.0%, Female: 22.3%, n = 7,532, $\chi^2$ = 71.561, p < 0.0001).

The gender disparity in pay was present even after male and female postdocs were matched in age, years since graduation, self-identification as white/Caucasian or other, satisfaction with mentorship, Carnegie classification of institution, public or private control of institution, whether married/partnered, or having children (nominal logistic regression model, gender effect test n = 7,280, $\chi^2$ = 51.330, p < 0.0001; *Supplementary file 1* (Table S4)). In other words, female postdocs earned less than male postdocs, regardless of type of institution, marital status, parental status, or majority/minority status. This gender wage gap increased with postdoc age but not with partnership status, partially supporting previous analyses of the STEM gender wage gap (*Association of American University Professors, 2006*; *Athanasiadou et al., 2018*).

Primary field of study was excluded from this analysis because field has such a large effect on salary, overshadowing most other factors, with postdocs in Engineering, Environmental Sciences, the Physical Sciences, and the Social Sciences earning significantly more money than postdocs in the Life Sciences, Humanities, Medicine, or Psychology (n = 7,542, 750.452, p < 0.0001; *Supplementary file 1* (Table S5)). Carnegie classification also had a large effect on salary (*Supplementary file 1* (Table S4)), as 58% of the postdocs at national government laboratories report earning more than $55,000 a year, while only 8% of postdocs at R1 institutions report earning more than $55,000 a year. Male postdocs were more likely than women to be in the primary fields of Engineering (n = 620, $\chi^2$ = 76.652, p < 0.0001) or Physical Sciences (n = 846, $\chi^2$ = 77.466, p < 0.0001), two fields which have historically higher salaries (*Buffington et al., 2016*). Interestingly, female postdocs trended towards being paid less than men in all fields except the Physical Sciences, where women trended towards being paid slightly more than men (*Supplementary file 1* (Table S5)). Income, mentor gender, citizenship, and partner status are all factors that may contribute to the observed gender difference in interest in primarily research-focused academic careers (*Buffington et al., 2016*) (*Figures 1* and *3H*).

Most postdocs reported salaries in the range of $39,000–$55,000 (median $43,750, mean $46,988, n = 7,551). In the 2014 National Postdoctoral Association's Institutional Policy Report, 52% of the 74 institutions reported that their minimum stipend matched the current NIH NRSA minimum (*Ferguson et al., 2014*). At the time of this survey, the NIH minimum was $43,692 (*National Institutes of Health, 2018*), which matches well with $43,750, the reported median income in our study, as well as the median postdoc income reported in recent work by McDowell et al. (*McDowell et al., 2018*). Five percent of postdocs reported mean gross incomes of less than $39,000 and ~10% reported incomes above $55,000. Although salaries in high cost of living (COL) urban areas tend to be higher than average (*Supplementary file 1* (Table S6)), when adjusted to publicly available COL data, postdocs in large metropolitan areas earn significantly less money than postdocs in college towns or rural settings (average salary when adjusted for cost of living, metropolitan: $38,045.60; non-metropolitan: $44,714.40;

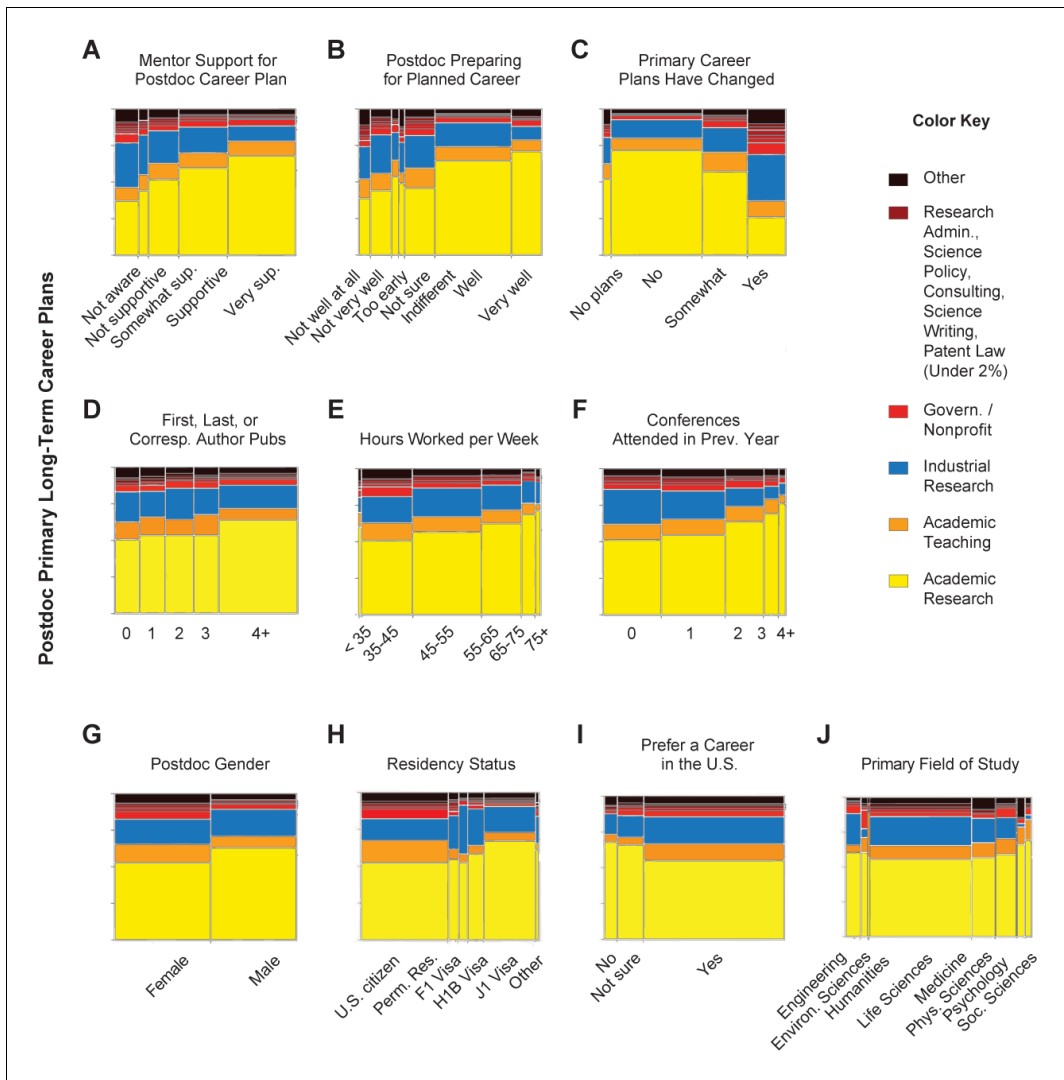

**Figure 3.** Postdoc career choice. Illustration of the independent effects of 10 of the 14 significant factors (out of 26) in the nominal logistic regression model of best fit for postdoc primary career choice (See *Table 1* for effect statistics). A–C illustrate the effect of postdoc mentor and postdoc confidence on postdoc career choice; D–F illustrate the effect of postdoc productivity on postdoc career choice, and G–J illustrate the effect of demographics on postdoc career choice. In these mosaic plots, the panels show the listed factor and corresponding effect size, and the right-hand color key corresponds to primary career choice. Factors are paraphrased survey questions; please see *Source Data 1* and *Source Data 2* for specific wording of questions.
DOI: https://doi.org/10.7554/eLife.40189.004

The following figure supplement is available for figure 3:

**Figure supplement 1.** Additional significant factors influencing career choice, but not depicted in *Figure 3*.
DOI: https://doi.org/10.7554/eLife.40189.005

n = 7,551, F-ratio:12.614, p = 0.0002; *Figure 2A*, *Supplementary file 1* (Table S6)). 'The Postdoctoral Experience Revisited' 2014 report recommended as a best practice that the minimum salary be set at $50,000 *(National Academy of Sciences, 2014)*; however, this has not been enacted at most institutions, and was only enacted by the NIH in November 2018. During the months that our survey was open (February–September 2016), the effect of a proposed minimum salary update ($47,476) to the Fair Labor Standards Act (FLSA) on postdoctoral salaries was openly debated, but ultimately not federally mandated (*Benderly, 2016*). Our data suggest that setting a minimum salary for postdocs is particularly important for postdoctoral researchers in large

metropolitan areas, where salaries are not maintaining parity with cost of living increases.

The majority of postdocs selected research-focused academic careers as their primary long-term career plan (57.7%), with industry research a distant second (17.8%; *Figure 1J*; *Supplementary file 1* (Table S2)). Determining the 'why' of career choice remains the subject of much study (*National Academy of Sciences, 2014*; *The National Academies of Sciences, 2018*; *National Institutes of Health, 2012*; *Hayter and Parker, 2018*; *Roach and Sauermann, 2017*). To assess which factors were most influential for determining postdoc career plan in our dataset (categorized in this survey as: academia, primarily research based; academia, primarily teaching based; industry; government/non-profit; other) we conducted a nominal logistic regression model with 26 factors concerning topics considered to be important for postdoc success and career choice (*Supplementary file 1* (Table S7)), which include demographics, training, productivity and mentor support matrices. The 14 significant factors in the model were (in order of effect size): 1) whether postdoc career plans had changed; 2) whether the postdoc received training in pedagogy; 3) feelings of career preparedness; 4) perceived support of career plan from mentors; 5) primary field of study; 6) residency status in the US; 7) intensity of job search; 8) postdoc gender; 9) number of

first, last, or corresponding author publications; 10) number of conferences attended in the past year; 11) hours worked per week; 12) total number of publications while a postdoc; 13) mentor rank; and 14) desire to pursue a career in the US (*Table 1*).

Perceived mentor support, number of postdoc publications, hours worked per week, conferences attended, and postdoc feelings of career preparedness were all positively correlated with a choice to pursue a research-focused academic career (*Figure 3A,B,D–F*; *Figure 3— figure supplement 1C*). Male postdocs, and postdocs who were not US citizens, were more interested in academic research positions (*Figure 3G and H*). In contrast, postdocs with mentors outside of the professoriate were more likely to prefer government/non-profit positions (*Figure 3—figure supplement 1A*). Whether this is a cause or effect relationship is not clear from our study, though we did find that postdocs with non-academic mentors changed their career plans at the same rate as those with academic mentors (n = 7,361, $\chi^2$ = 6.860, p = 0.077). In addition, postdocs actively searching for permanent positions were less interested in academic research than postdocs not yet on the job market (*Figure 3—figure supplement 1B*), and were more likely to have changed their career plans (n = 7,565, $\chi^2$ = 224.633, p < 0.0001). These results complement recent

**Table 1.** Significant factors influencing postdoc primary career plans.

| Factor | $\chi^2$ | -log p-value |
|---|---|---|
| Whether long-term career plans have changed | 599.951 | 108.529 |
| Received training in pedagogy | 151.052 | 27.273 |
| Feelings of career preparedness | 161.510 | 11.925 |
| Perceived mentor support of career plan | 130.577 | 11.925 |
| Primary field of study | 191.331 | 10.190 |
| Residency status in US | 133.264 | 9.941 |
| Job search intensity | 98.574 | 9.352 |
| Postdoc gender | 53.654 | 7.658 |
| Number of first, last, or corresponding author publications | 86.193 | 5.274 |
| Conferences attended in last year | 84.468 | 5.043 |
| Hours worked/week | 109.093 | 4.870 |
| Total number of publications while a postdoc | 80.503 | 4.524 |
| Academic rank of mentor | 70.513 | 3.292 |
| Plan to pursue a career in US | 37.452 | 2.340 |

A nominal logistic regression model that considered 26 factors that might be important for postdoc success and career choice revealed 14 significant factors. Factors are listed in order of decreasing effect size. Nominal logistic regression model, whole model effect: n = 6,504, Model $R^2$ = 0.2017, AICc = 15924, BIC = 21130.

DOI: https://doi.org/10.7554/eLife.40189.006

studies suggesting that individual career choice is influenced by changing job attribute preferences and self-awareness (*Buffington et al., 2016*), and that academic success is influenced by mentorship during the postdoctoral period (*Lienard et al., 2018*).

Sixty percent of respondents were either satisfied or very satisfied with the mentorship they receive, with similar responses from both genders (*Figure 1I*). To assess which factors were most influential for determining satisfaction with their postdoc mentor, we conducted a nominal logistic regression model with the same 26 factors included in the model for postdoc career choice (though excluding satisfaction with mentorship as a factor, and replacing it with postdoc long-term career plan; *Supplementary file 1* (Table S7)). The eight significant factors in the model (in order of effect size) were: 1) feelings of career preparedness; 2) perceived support of career plan from mentors; 3) frequency of project meetings with mentor; 4) intensity of job search; 5) whether the postdoc received training in mentorship; 6) primary field of study; 7) perception of job market; and 8) academic rank of mentor (*Table 2*). These factors were more important than number of postdoc publications, whether a postdoc had changed career plans, postdoc or mentor gender, residency status, or postdoc training in either grant writing or pedagogy.

Perceived mentor support had a positive effect on how satisfied a postdoc is with their mentor, as did frequency of mentor meetings, perception of preparedness for desired future career, and perception of job market (*Figure 4A–D*). Postdocs who received training in mentorship were more satisfied with the mentorship they received than postdocs who did not receive training in mentorship (*Figure 4E*). We found this to be particularly noteworthy, as mentorship training is not a common part of the postdoctoral experience, with only 26% of postdocs reporting that they have received such training. While we cannot comment further on the specific type of mentor training that postdoc respondents received, we note that several institutions in our study have mentor training programs for postdocs in place, including those that use curricula from the University of Wisconsin-Madison Center for the Improvement of Mentored Experiences in Research (CIMER) project and/or National Research Mentoring Network (NRMN), for example the Big Ten Alliance NRMN-CAN program, as well as the 'Mentoring in Research' program at Stanford University.

Previous research on a postdoc cohort showed that high satisfaction with mentorship and perceived support correlated with increased interest in an academic research focused career (*Scaffidi and Berman, 2011*). In addition, in a randomized, controlled study, a different type of mentoring, 'group career coaching,' where a career coach works with small groups of graduate students to provide support and promote career progress, was found to increase both perceived 'achievability' and 'desirability' of academic careers in an under-represented minority student group (*Williams et al., 2016*). Thirdly, in a longitudinal study of PhD students interested in academic careers, their perceived ability, or self-efficacy, was a strong indicator of retaining interest in a faculty career (*Roach and Sauermann, 2017*).

**Table 2.** Significant factors influencing postdoc satisfaction with mentoring.

| Factor | $\chi^2$ | -log p-value |
|---|---|---|
| Feelings of career preparedness | 960.457 | 181.948 |
| Perceived mentor support of career plan | 904.891 | 178.146 |
| Frequency of mentor meetings | 532.31 | 89.480 |
| Job search intensity | 68.255 | 8.040 |
| Received training in mentorship | 37.088 | 6.240 |
| Primary field of study | 92.193 | 4.368 |
| Perception of academic job market | 48.088 | 3.384 |
| Academic rank of mentor | 41.614 | 2.508 |

A nominal logistic regression model was calculated based on the same 26 factors used to model postdoc primary career plans (*Table 1*). Eight of these factors were found to be significant; factors are listed in order of decreasing effect size. Nominal logistic regression model, whole model effect: n = 6,504, Model $R^2$ = 0.3007, AICc = 14729, BICc = 17810.

DOI: https://doi.org/10.7554/eLife.40189.007

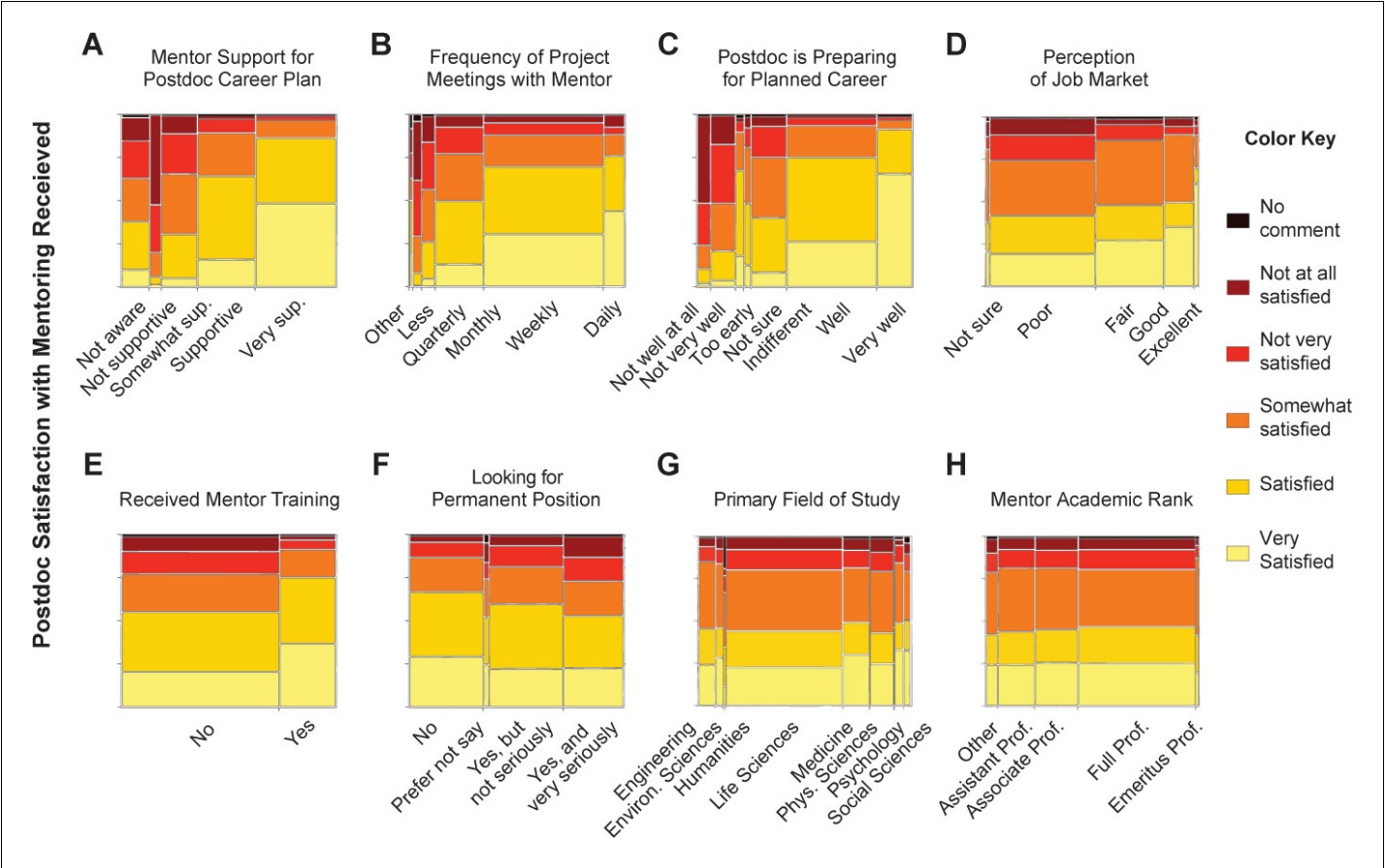

**Figure 4.** Postdoc satisfaction with mentor. Illustration of the independent effects of the eight significant factors (out of 26) in the nominal logistic regression model of best fit for satisfaction with postdoc mentor (see *Table 2* for effect statistics). In these mosaic plots, the panels show the listed factor and corresponding effect size, and the right-hand color key corresponds to the degree of satisfaction with mentor. Factors are paraphrased survey questions; see *Source Data 1* and *Source Data 2* for specific wording of questions.

DOI: https://doi.org/10.7554/eLife.40189.008

Other studies suggest that in addition to structured oversight and professional development (*Davis, 2009*), provision of research mentor training improves the success of researchers-in-training at all levels (*Fleming et al., 2012*). These studies strongly suggest that increasing a graduate student or postdoc's self confidence and self-efficacy increases interest in an academic faculty career. Self-efficacy is directly impacted by the primary mentor and may explain the correlation between perceived support from their mentor and a postdoc's interest in a faculty career seen here and elsewhere (*Scaffidi and Berman, 2011*). While our data do not show a significant correlation between gender and satisfaction with their mentor, they do suggest that an increase in mentor support and mentorship training will increase a postdoc's interest in academic jobs. This increase in mentor support and mentorship may be a

particularly important tool for increasing female and under-represented postdocs' pursuit of research-intensive academic careers.

## Conclusions

In summary, our dataset represents the most comprehensive survey of the US postdoctoral population in over a decade. As such, these data may provide a benchmark for legislation and institutional policy makers, inform research questions pertaining to the evolving postdoctoral population, and serve as a precedent for understanding the important dynamics of the scientific workforce.

We found that a research-focused academic position remains the most common primary career goal for postdocs, in spite of increasing emphasis on other types of careers for doctorate holders (*The National Academies of Sciences, 2018*; *Alberts et al., 2014*; *St Clair et al.,*

*2017*). Although 60% of respondents were satisfied with the mentoring that they receive, our data suggest that providing formal mentorship training for postdocs may significantly increase their satisfaction with their mentor and influence career choice (*Williams et al., 2016*). Our data also show that women are less interested in research-focused academic positions than men, and this may be associated with gender specific differences in postdoctoral experiences (*Moss-Racusin et al., 2012*).

While the data we collected allowed us to identify a number of factors influencing the postdoctoral experience, other factors, such as socioeconomic background and underrepresented status, may also play a significant role, and should be studied further. Nevertheless, our findings highlight the impact of mentoring, across all demographics, as essential to informing career choice and determining quality of postdoctoral experience.

## Materials and Methods

### Survey instrument design

The National Postdoc Survey questions were designed to emphasize aspects of the postdoctoral experience related to career choice and mentoring, in addition to collecting basic demographic data. These questions were based on over a decade of experience with postdoctoral surveys administered at the University of Chicago, led by postdocs within the Biological Sciences Division Postdoctoral Association. In an effort to maximize participation for all postdocs, regardless of institutional environment, we disseminated the survey using top-down and grass-roots methods described below.

We conducted the survey in two phases: a 15-institution pilot phase, followed by a national rollout to over 450 institutions. The pilot phase was launched on February 2, 2016 after contacting and inviting participation from administrators at the 15 member schools of the Committee on Institutional Cooperation (CIC, now the Big Ten Academic Alliance plus the University of Chicago). 272 postdocs participated in the pilot phase of the survey. Feedback about the survey design was solicited during a workshop about the survey presented at the National Postdoc Association Annual meeting on March 4, 2016. The pilot survey questions (*Source Data 1*) were then slightly modified before nationwide launch on March 31, 2016. These revisions included additional demographic questions, and

rephrasing of several questions to improve clarity (*Source Data 2* and *Source Data 3*). The revised survey was available from March 31–September 2, 2016. While the CIC institutions participated in the pilot version of this survey, the survey was also open to postdocs at CIC institutions after the national rollout. A majority of participants from CIC institutions responded after March 31, 2016 and took the final version of the survey rather than the pilot version.

For the top-down survey dissemination approach, a team of five postdocs and two administrators compiled contacts for all doctoral degree and research institutions in the US that were thought to have postdoctoral researchers. We gathered publicly available contact information for Postdoctoral Offices, Postdoctoral Associations, as well as Offices of Research, Deans of Graduate Schools, Provosts and any other administrators that may represent postdocs for each institution (including website, email addresses and names) via web search. Whenever an institution did not have a postdoctoral office, we tried to determine who had oversight of postdoctoral researchers, such as a representative from an Office of Research, Graduate School, or a Provost Office. We used this information to simultaneously contact those who we determined were most likely to represent postdocs at each university, including any listed postdoc contacts. Multi-respondent emails were sent to the above described representatives at each institution. These individuals were again invited to participate during the months of April, June, July and August, and contact lists were revised to update contact information, and include additional institutions expected to have postdocs.

For our grass-roots survey dissemination approach we launched a website that could be freely shared on social media and by email, which explained the survey aims and contained a centralized contact form. The contact form allowed any postdocs who had not been reached via standard institutional contacts to participate in the survey through this secondary means of contact. In addition, we periodically checked contact information for institutional representatives, and updated the contact information, added new institutional contacts, and encouraged grass-roots survey dissemination during the seven months that the survey was active.

In all, 482 sets of putative postdoctoral oversight representatives were contacted by email, although some larger institutions such as Harvard University and NIH often housed separate

institutes or offices that were each contacted separately – in these cases five and 30 sets, respectively. During the seven months (February 2–September 2, 2016) that the survey was open, over 7,600 postdoc responses were collected, with respondents from every state, and from 351 institutions and universities. While the number of respondents varied between months (ranging from 24 during the two days the survey was open in September to 2,268 in August), there was no statistical difference in the gender ratio of respondents over the seven months (whole dataset: 53.1% female and 46.9% male ±5, n = 7,579, $\chi^2$ = 10.703, p = 0.1521; excluding non-US postdocs: 53.1% female and 46.9% male ±5; F:M, n = 7,560, $\chi^2$ = 10.866, p = 0.1446). Respondents from the 46 institutions that participated in the 2005 Sigma Xi survey, representing institutions with long-standing institutional support for postdocs, contributed 3,126 responses, slightly less than half of all respondents. This indicates that our addition of a grass-roots approach to survey dissemination contributed to a broader sampling of postdocs across different institutional environments, providing an even more comprehensive assessment of US postdoctoral experiences.

Four institution classifications were added as fixed variables to the final dataset: institution classification as public or private; Carnegie classification; US Census Region; and participation in the 2005 Sigma Xi Postdoctoral Survey. City and state of each institution were also added.

### Statistical analysis

Raw response data were quality-filtered to select for US-based institutions and individuals who were currently in self-described postdoctoral positions. Of the 7,673 total respondents, 70 were removed from the initial dataset using these quality filters, yielding a final dataset of 7,603 US postdoctoral respondents. The demographics data shown (*Figure 1*) were calculated by first sorting by gender, and then sorting by the demographic of interest displayed as total percentage of respondents per gender (all panels except *Figure 1H*) or by a mean ± standard deviation (*Figure 1H*) using Prism7 (GraphPad). The effect of gender on salary, having a same sex mentor, residency status, partner status, and having children was tested using a Pearson $\chi^2$ test (n = 7,516, 7,459, 7,543, 7,538, 7,532 respectively). Sample sizes differed because respondents were allowed to skip questions, and are therefore reported as 'n' here and throughout. However, most respondents

answered most survey questions, as can been seen by the similar sample sizes for these different survey questions. The effects of gender, age, years since graduation, satisfaction with mentor, and likelihood of being partnered on postdoc salary were tested using a nominal logistic regression model, n = 7,311. All survey questions reported here had categorical response options, thus we used nominal logistic regression models instead of generalized linear regression models to account for the categorical nature of the data. The effect of gender on being in the fields of engineering or the physical sciences was tested using a Pearson $\chi^2$ test, engineering n = 620, physical sciences n = 846. We used a Bonferroni correction to account for multiple testing, yielding a significance threshold of p = 0.006. All statistical tests were two-sided. Statistics were performed using JMP 13.1 by SASS.

To determine what factors were significantly correlated with postdoctoral career choice and satisfaction with mentor, we ran a nominal logistic regression model using 26 different fixed variables listed in *Supplementary file 1* (Table S7) using the JMP 13.1 by SASS fit model platform. We then determined which factors were significant variables after controlling for multiple testing. These estimates of effect size are reported in *Table 1* and *Table 2*. A total of 6,504 respondents answered all 26 of the questions included in this analysis.

### Cost of living and postdoc salaries

Cost of living index (COL) data for 2016 was produced by the Council for Community and Economic Research (https://www.c2er.org/). State COL data were generated by a weighted average across cities that have 2016 C2ER cost of living data provided per state, for the cities where postdoc salary data were available. Average postdoc salary from all survey respondents for each location was divided by matched local COL values to produce postdoc salaries adjusted by cost of living. Whenever income was not specified, the midpoint of income range selected by the respondent was used. These values were mapped to each state with red to blue corresponding to lowest to highest adjusted salary, respectively. In addition, counties with institutions having at least 50 respondents were then mapped separately, to map adjusted postdoc salary in 48 counties with additional COL data, against the background of the state COL data, in 50 states plus Washington DC.

## Population proportion analysis

To determine the number of individual responses required from a total population of 100,000 for 95% and 99% confidence levels, at a 5% margin of error, assuming the true population proportion being measured is between 3–50% of the total population, we conducted a population proportion analysis using the equation and definitions as described in Tintle et al. (*Tintle et al., 2016*) and at Select Statistical Services Limited (*Select Statistical Services Limited, 2018*). Results are reported in *Supplementary file 1* (Table S1).

## Data analysis of survey respondent proportions

Inconsistent definitions across institutions and lack of existing institutional contact lists for postdocs, particularly for those without postdoctoral offices and other support, can make collecting representative data for postdocs challenging (*Schaller et al., 2017*). Thus demographics of respondents may differ across surveys, and the postdoctoral demographics of previous survey datasets may differ from those observed in our study. To further assess our demographic data, we conducted the comparisons described below.

We compared our demographics to that of the 2005 Sigma Xi survey, which is perhaps the most comparable effort to our own, having 7,600 postdoc respondents, both citizens and noncitizens (*Davis, 2005*). The 2005 Sigma Xi dataset had 42% female postdocs (51% female for US citizens, and 35% female for internationals), and overall 46% US citizen and permanent resident postdocs (54% temp visas). Our current survey dataset contains a higher percentage of both female postdocs (53% female) and US citizen and permanent resident postdocs (55%) relative to the Sigma Xi survey from a decade ago, which may in part reflect changing demographics of the US postdoctoral population, as well difference in institutions sampled. However, the relative difference in proportion of females for US and non-US citizens remains consistent (approximately 15%); our US citizen respondents were 60% female, while our international respondents were 46% female.

An alternative explanation for this increase in female respondents in our dataset relative to the earlier Sigma Xi survey is that women may have disproportionately responded to our survey. We tested this hypothesis by checking the University of Chicago female and male response ratio against the actual sex ratio of female and male postdocs in the Biological Sciences. Our survey respondents were 49.3% female and 50.7% male, while the actual sex ratio of female and male postdocs in the University of Chicago Biological Sciences was 46.5% female and 53.5% male, which puts our survey respondent ratio well within the standard 5% margin of error. While it is unclear how representative University of Chicago postdocs are of the national postdoctoral population, it is important to remember that the surveyed population, by definition, all have advanced degrees, work at research institutions, and are all highly likely to have strong command of the English language, even if it is not their first language. Doctorate recipients make up 2% of the US national population (*United States Census Bureau, 2015*). As doctorates are a small percentage of the national population, they are likely to make up a small percentage of respondents to general national surveys. Thus response biases of surveys targeting this population may differ from those targeting the general population.

## Acknowledgements

We thank Heather Titley and Giorgio Grasselli for assistance with survey instrument design and dissemination; Laurie Risner for assistance with the Big Ten Pilot Phase; and Dylan Meyer for assistance with data clean-up. We express deep gratitude to all postdocs who participated in this survey, as well as to the postdoctoral associations, administrators, and many others who helped disseminate our survey.

**Sean C McConnell** is in the Department of Science, Medicine and Public Health, American Medical Association, Chicago, United States

**Erica L Westerman** is in the Department of Biological Sciences, University of Arkansas, Fayetteville, United States

ewesterm@uark.edu

https://orcid.org/0000-0002-3575-8298

**Joseph F Pierre** is in the Department of Pediatrics and Department of Microbiology, Immunity, and Biochemistry, University of Tennessee Health Science Center, Memphis, United States

**Erin J Heckler** is in the Office of Postdoc Affairs in the Office of the Vice Chancellor for Research, Washington University in St. Louis, St. Louis, United States

**Nancy B Schwartz** is in the Department of Pediatrics and Department of Biochemistry and Molecular Biology, University of Chicago, Chicago, United States

**Author contributions:** Sean C McConnell, Joseph F Pierre, Conceptualization, Formal analysis, Visualization, Methodology, Writing—original draft, Project administration, Writing—review and editing, Designed the survey and analyzed data, Disseminated the survey; Erica L Westerman, Conceptualization, Formal analysis, Visualization, Methodology, Writing—original draft, Project administration, Writing—review and editing, Designed the survey and analyzed data, Performed multivariate analysis and models, Disseminated the survey; Erin J Heckler, Conceptualization, Visualization, Methodology, Writing—original draft, Project administration, Writing—review and editing, Designed the survey and analyzed data, Disseminated the survey; Nancy B Schwartz, Writing—original draft, Project administration, Writing—review and editing

**Competing interests:** The authors declare that no competing interests exist.

**Ethics:** Human subjects: Participation in this survey was completely voluntary. In the introduction to this survey, we informed the participants of its purpose, and that results of the survey would be disseminated, in aggregate. All responses were recorded in a secure RedCap Database, so they could not be traced back to individual respondents. Responses were combined for data analysis to maintain respondent anonymity throughout data analysis. Our survey design and dissemination protocol was approved by the University of Chicago Institutional Review Board, IRB Protocol Number 15-1724.

### Funding

| Funder | Grant reference number | Author |
|---|---|---|
| Biological Sciences Division at the University of Chicago | | Nancy B Schwartz |
| Institute for Translational Medicine and Therapeutics | CTSA NIH UL1 TR000430 | Nancy B Schwartz |
| University of Chicago Biological Sciences Division Postdoctoral Association and Office of Graduate and Postdoctoral Affairs | | Erin J Heckler Nancy B Schwartz |
| National Research Mentoring Network-Committee on Institutional Cooperation Academic Network (NRMN-CAN) sub award 5101964-6 | | Nancy B Schwartz |

At time of project inception, all authors were either members of the University of Chicago Biological Sciences Division Postdoctoral Association, the Assistant Director of Postdoctoral Affairs, or the Dean of Postdoctoral Affairs at the University of Chicago. Additional funders had no role in study design, data collection and interpretation, or the decision to submit the work for publication.

## Additional files

### Supplementary files

• Source data 1. Dataset 1: Pilot National Postdoc Survey used Feb 2–March 30, 2016
DOI: https://doi.org/10.7554/eLife.40189.009

• Source data 2. Dataset 2: National Postdoc Survey used March 31–Sept 2, 2016
DOI: https://doi.org/10.7554/eLife.40189.010

• Source data 3. Dataset 3: Changes between pilot and final survey
DOI: https://doi.org/10.7554/eLife.40189.011

• Supplementary file 1. Tables S1 to S7. Table S1. Population proportion analysis. Number of individual responses required from a total population of 100,000 for 95% and 99% confidence levels, at a 5% margin of error, assuming the true population proportion being measured is between 3% and 50% of the total population. Response values estimated using a population proportion analysis following equations and definitions described in Tintle et al. (*Davis, 2009*) and at Select Statistical Services Limited (*Fleming et al., 2012*). Table S2. National data summarized in main text and *Figure 1*. Table S3. Percentages of postdoc respondents in primary fields per US census region. There was a small, but significant, correlation between region and field (Pearson $\chi^2$, n = 7,585, $\chi^2$ = 134.145, p < 0.0001). Table S4. Nominal logistic model of gender disparity in pay. Gender remains a significant factor explaining postdoc salary, even when including year of terminal degree, age, partner status, parental status, type of institution, institution control, and satisfaction with mentor. Whole model n = 7,280, $\chi^2$ = 2589.077, p < 0.0001, AICc = 28458.9, BIC = 30056.2. Table S5. Gender salary disparity

by field. Table S6. Respondent reported salaries adjusted to cost of living. Table S7. Factors included in nominal logistic regression models of satisfaction with mentor and primary career plan.
DOI: https://doi.org/10.7554/eLife.40189.012

• Transparent reporting form
DOI: https://doi.org/10.7554/eLife.40189.013

## Data availability

Non-privileged data used in this study are available in supplemental tables and additional material related to this manuscript. Due to their sensitive nature, much of the raw data are privileged to prevent individual identification, in accordance with IRB protocol. However, summary data for institutions, fields, and regions with more than 50 respondents are available upon request.

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
