## [Decision Letter]

Thank you for submitting your article "United States National Postdoc Survey Results and the Interaction of Gender, Career Choice, and Mentor Impact" for consideration by *eLife*. Your article has been reviewed by two peer reviewers, and the evaluation has been overseen by Emma Pewsey as the Reviewing Editor and Peter Rodgers as the Senior Editor. The following individual involved in the review of your submission has agreed to reveal their identity: Jennifer Miller (Reviewer #1).

The reviewers have discussed the reviews with one another and I have drafted this decision to help you prepare a revised submission.

Summary

This study is the first nationally comprehensive survey of postdocs in over a decade. For far too long, institutions, government labs and industry teams have not kept consistent or reliable data on postdocs or equivalent positions. The study addresses a significant void and is a first step towards better understanding the experience of postdocs and how different factors impact their job satisfaction and their future career decisions. The authors should also be applauded for tying their salary recommendations to cost of living. This will be important for political feasibility, and it is also consistent with the data.

Essential revisions

1. The paper would integrate more readily with other studies of the scientific workforce if data were analyzed using regression analysis. The main advantage of a regression analysis in this case would be building a model with causal rather than descriptive framing. If you choose not to perform a regression analysis, please explain in the text why you have chosen to use chi-squared tests instead.

2. Given the importance of postdoc mentor training to the article, a more robust discussion on it is warranted. What is considered to be 'training'? Please provide more information on the goals of postdoc mentor training, the nature of mentor training or the effectiveness of the training. Please also make it clear whether only the mentors of the postdocs undergo training, or whether, in some instances, the postdocs themselves also do so.

3. The correlation between perceived mentor support and the interest in a faculty career was highly interesting, and the article would benefit from a more in-depth discussion about why this was possibly the case.

4. This study did a very good job of recruiting participants from a wide range of institutions across the country, but it would help to know more about the nature of the scientific environments that the respondents are currently in – what type of institution, and what level of funding the institutions typically hold. This sort of information would help to paint a clearer picture of the respondents and would help to better contextualize the diversity of the pool.

5. There is significant discussion about the gender gap for postdoc pay equity. It would be helpful to provide more information about the individuals experiencing the pay inequity to understand the nature of the disparity.

---

## [Author Response]

1. The paper would integrate more readily with other studies of the scientific workforce if data were analyzed using regression analysis. The main advantage of a regression analysis in this case would be building a model with causal rather than descriptive framing. If you choose not to perform a regression analysis, please explain in the text why you have chosen to use chi-squared tests instead.

We agree with the reviewers that regression analyses are most appropriate for these data. That is why we used regression models throughout the study. Specifically, we used nominal logistic regression models, to take into account the categorical nature of the responses and factors. We now clarify this in the text. The test statistics for nominal logistic regression models are χ^2^, which we report in addition to p-values for whole models and effect tests. We now clarify this in both our Methods and Results sections.

2. Given the importance of postdoc mentor training to the article, a more robust discussion on it is warranted. What is considered to be 'training'? Please provide more information on the goals of postdoc mentor training, the nature of mentor training or the effectiveness of the training. Please also make it clear whether only the mentors of the postdocs undergo training, or whether, in some instances, the postdocs themselves also do so.

To clarify, we only refer to postdocs undergoing training in mentorship themselves, not their faculty mentors. We did not ask if the postdoc’s mentor had received training.

We allowed respondents to self-define their answer to the question, “Have you received mentor training?”. Thus we are unable to answer questions on the content, effectiveness, or goals of mentor training from the survey data. Of the postdoc respondents who selected “Yes”; 36% indicated in a later matrix question that the “majority of your training in [mentoring] comes from” their primary faculty mentor / PI and did not select other options. The other options (in order of most often selected) were self, other faculty, other combination, collaborators, and lab members in that order. The majority of respondents chose some combination of offered selections.

That being said, there are mentor training programs for postdocs at several institutions included in this study. The NRMN-CAN Big Ten Alliance professional development program, the UW-Madison Wisconsin CIMER Project and NRMN curricula, and “Mentoring in Research” at Stanford University are examples of such programs.

We have now revised the text (Results and Discussion section) to read:

“Postdocs who received training in mentorship were more satisfied with the mentorship they themselves received than postdocs who did not receive training in mentorship (Figure 4E). We found this to be particularly noteworthy, as mentorship training is not a common part of the postdoctoral experience, with only 26% of postdocs reporting that they have received such training. While we cannot comment further on the specific type of mentor training that postdoc respondents received, we note that several institutions included in our study have mentor training programs for postdocs in place, including those that use curricula from the UW-Madison CIMER project and/or National Research Mentoring Network (NRMN), e.g. Big Ten Alliance NRMN-CAN program, as well as “Mentoring in Research” at Stanford University.”

3. The correlation between perceived mentor support and the interest in a faculty career was highly interesting, and the article would benefit from a more in-depth discussion about why this was possibly the case.

We have now expanded our discussion of the potential causes of the correlation between perceived mentor support and interest in a faculty career, and include the following sentences (Results and Discussion section):

“Thirdly, in a longitudinal study of PhD student interest in academic careers, their perceived ability, or self-efficacy was a strong indicator of retaining interest in a faculty career (Ref 25). Taking these studies as a whole, high self-efficacy reinforces interest in an academic faculty career. In turn, self-efficacy is directly impacted by the primary mentor and may explain the correlation between perceived support from their mentor and a postdoc’s interest in a faculty career seen here and in Ref 27.”

4. This study did a very good job of recruiting participants from a wide range of institutions across the country, but it would help to know more about the nature of the scientific environments that the respondents are currently in – what type of institution, and what level of funding the institutions typically hold. This sort of information would help to paint a clearer picture of the respondents and would help to better contextualize the diversity of the pool.

We now include information on how many of our respondents were from R1 academic institutions, non-R1 academic institutions, medical centers, government labs, and other, as well as the proportion of postdocs in academia who were located at private versus public institutions.

*5. There is significant discussion about the gender gap for postdoc pay equity. It would be helpful to provide more information about the individuals experiencing the pay inequity to understand the nature of the disparity*.

We now expand on gender disparity in salary, and clarify that this salary disparity was independent of where postdocs worked, marital status, parental status, age, or race/ethnicity. As far as we can tell, nearly all groups of female postdocs, with the exception of those in the physical sciences, earn less than their male counterparts.